# Therapeutic Outcomes and Prognostic Factors of Unresectable Intrahepatic Cholangiocarcinoma: A Data Mining Analysis

**DOI:** 10.3390/jcm10050987

**Published:** 2021-03-02

**Authors:** Tomotake Shirono, Takashi Niizeki, Hideki Iwamoto, Shigeo Shimose, Hiroyuki Suzuki, Takumi Kawaguchi, Naoki Kamachi, Yu Noda, Shusuke Okamura, Masahito Nakano, Ryoko Kuromatu, Hironori Koga, Takuji Torimura

**Affiliations:** Division of Gastroenterology, Department of Medicine, School of Medicine, Kurume University, Kurume 830-0011, Japan; niizeki_takashi@kurume-u.ac.jp (T.N.); iwamoto_hideki@med.kurume-u.ac.jp (H.I.); shimose_shigeo@med.kurume-u.ac.jp (S.S.); suzuki_hiroyuki@med.kurume-u.ac.jp (H.S.); takumi@med.kurume-u.ac.jp (T.K.); kamachi_naoki@med.kurume-u.ac.jp (N.K.); noda_yuu@med.kurume-u.ac.jp (Y.N.); okamura_shyuusuke@kurume-u.ac.jp (S.O.); nakano_masahito@kurume-u.ac.jp (M.N.); ryoko@med.kurume-u.ac.jp (R.K.); hirokoga@med.kurume-u.ac.jp (H.K.); tori@med.kurume-u.ac.jp (T.T.)

**Keywords:** cholangiocarcinoma, prognostic variable, inflammation, interventional radiology, reservoir system

## Abstract

Prognosis of patients with intrahepatic cholangiocarcinoma (ICC) is unsatisfactory. Tumor, host, and treatment factors including hepatic arterial infusion chemotherapy (HAIC) are intricately involved in the progression of ICC. We aimed to identify profiles associated with disease control rate (DCR) and the prognosis of patients with unresectable ICC by decision tree analysis. We analyzed 31 consecutive patients with unresectable ICC (median age, 71 years; the male ratio was 58.1%). Stage IVB occupied 51.6% of patients, and 38.7% and 58.1% of patients were treated with gemcitabine plus cisplatin combination therapy and HAIC, respectively. Profiles associated with prognosis as well as DCR were investigated by decision tree analysis. The median survival time (MST) of the patients was 11.6 months, and the DCR was 70.9%. Multivariate correlation analysis showed that albumin levels and WBC levels were significantly correlated with survival time (albumin, ρ = 0.3572, *p* = 0.0485; WBC, ρ = −0.4008, *p* = 0.0280). In decision tree analysis, WBC level was selected as the initial split variable, and subjects with WBC levels of 6800/μL or less (45.1%) showed a long survival time (MST 476 days). We also demonstrated that the profile associated with the highest DCR was “less than 4.46 mg/dL of CRP levels and treatment with HAIC”. We demonstrated a new prognostic profile for ICC patients, which consisted of WBC and CRP levels. Moreover, we demonstrated that HAIC was associated with better disease control in ICC patients with low CPR levels. Thus, these new profiles may be useful for the management of ICC patients.

## 1. Introduction

Cholangiocarcinoma is a malignant tumor that originates from the epithelium of the bile duct [1]. Depending on the site of the bile duct where it occurs, cholangiocarcinoma is classified as either extrahepatic cholangiocarcinoma or intrahepatic cholangiocarcinoma (ICC). The incidence of ICC has increased in recent decades and accounts for 4.4% of all liver tumors. ICC is the second-most common primary liver malignancy in Japan [2]. Although surgery is considered the only potentially curative treatment, the 5-year overall survival (OS) rate after surgery is 25–31% [3,4], and the recurrence rate is 40–64% [3,5,6]. In addition, the resectability rate is low because this disease is frequently beyond the limits of surgical therapy at the time of diagnosis [7]. Moreover, the median survival time (MST) was 10 months in patients who were treated with chemoradiotherapy [8]. A meta-analysis showed that median and 5-year OS generally are approximately 28 months (range, 9–53 months) and 30% (range, 5–56%), respectively [9]. Thus, the prognosis of patients with ICC is poor.

Regional therapy—including hepatic arterial infusion chemotherapy (HAIC)—is one treatment option for advanced ICC [10]. Compared with systemic chemotherapy, HAIC puts patients at risk for some adverse events, including arterial occlusion, subcutaneous hematomas, and infections. However, HAIC is able to deliver high concentrations of anticancer drugs to the liver with fewer systemic side effects [11,12]. Recently, a few studies have examined the effect of HAIC on the survival of patients with advanced ICC. Kasai et al. reported that the survival rates at 1 and 2 years were 53.7% and 14.3%, respectively, and the MST was 14.6 months by treatment with HAIC with 5-fluorouracil combined with subcutaneous administration of pegylated interferon α-2b in patients with advanced ICC.

The objective response rate (complete response (CR) + partial response (PR)/all cases) and disease control rate (DCR; CR + PR + stable disease (SD)/all cases) were 60.0% and 90.0%, respectively [13]. Konstantinidis et al. reported that survival rates at 1 and 2 years were more than 80% and more than 60%, respectively, while the disease control rate was 94% and the MST was 30.8 months by treatment with a combination of systemic chemotherapy and HAIC with floxuridine, mitomycin C, or gemcitabine [14]. Thus, HAIC seems to be associated with a high DCR; however, limited information is available for the DCR of HAIC in patients with advanced ICC.

The molecular targets in the treatment of ICC have not been identified so much. In the last few years, novel treatment targets have been identified in ICC patients, including fibroblast growth factor receptor (FGFR) aberrations; thus, several FGFR inhibitors are currently being developed, some of which have already suggested interesting efficacy and adequate safety in phase I and phase II trials regarding refractory ICC [15]. Inflammation is also an important finding in tumor progression. Several reports reveal that high inflammatory condition was correlated with poor prognosis of patients with advanced cancers. In the present study, we studied whether the inflammatory serum markers were associated with the prognosis of patients with advanced ICC. Data mining analysis is an artificial intelligence approach to reveal factors and interactions between variables from data sets, even if no a priori hypothesis has been imposed [16]. The benefits of this approach include the discovery of hidden profiles and the provision of additional information that cannot be identified through logistic regression analysis. Decision tree analysis is able to identify priorities used to reveal a series of classification rules [17,18], and the results can be used to make stepwise decisions about disease management [19].

Recently, decision tree analysis has been used to investigate prognostic factors for patients with pancreatic cancer [20], breast cancer [21], and leukemia [22]. In addition, Tsilimigras et al. recently reported patients’ characteristics associated with survival of patients with resectable ICC using decision tree analysis and showed four favorable characteristics, namely single ICC, size < 5 cm, albumin–bilirubin grade I, and negative preoperative lymph node status [23]. To our knowledge, however, decision tree analysis has never been used to investigate the profiles associated with the prognosis of patients with unresectable ICC.

The purpose of this study is to identify profiles associated with DCR and the prognosis of patients with unresectable ICC by decision tree analysis.

## 2. Materials and Methods

### 2.1. Study Design

This retrospective study aimed to identify profiles associated with the DCR and prognosis of patients with unresectable ICC by decision tree analysis. This protocol conformed to the ethical guidelines of the 1975 Declaration of Helsinki, as reflected by the prior approval of the institutional review board of Kurume University. All examinations and treatments were performed in accordance with relevant guidelines and regulations. An opt-out approach was used to obtain informed consent from patients, and personal information was protected during data collection.

### 2.2. Subjects

A total of 39 consecutive patients were diagnosed with ICC between 2008 and 2018 in our institution. Of these, eight patients who were treated with best supportive care (*n* = 5) and hepatic resection (*n* = 3) were excluded to identify the optimal treatment for unresectable ICC. Thus, the remaining 31 patients with ICC were used in our analysis (Figure 1).

### 2.3. Diagnosis of ICC

Of the 31 patients with ICC, 80.6% (25/31) were diagnosed via histological examination, and 19.4% (6/31) were diagnosed with a combination of serum tumor markers, such as carcinoembryonic antigen (CEA) and CA 19-9 antigen, as well as imaging modalities such as ultrasonography, computed tomography, magnetic resonance imaging, and/or angiography according to the Guidelines for the Diagnosis and Management of ICC [3].

### 2.4. Inclusion and Exclusion Criteria

The following patient inclusion criteria were used: (1) ICC, (2) age > 18 years, (3) no previous treatment for ICC, (4) treatment with systemic chemotherapy or HAIC, and (5) complete follow-up from the initial treatment for ICC until death or the study censor time (November 2018). The following patient exclusion criteria were used: (1) treatment with hepatic resection, (2) best supportive care, (3) history of a malignant tumor other than ICC within the 5 years preceding the study, and (4) participation in any drug trial.

### 2.5. Data Collection

Variables related to host, tumor, and treatment factors were retrospectively reviewed using clinical records. The following data were collected at the time of diagnosis of ICC before chemotherapy: host factors, including age, sex, white blood cell (WBC) count, neutrophil-to-lymphocyte ratio (NLR), hemoglobin level, platelet count, prothrombin activity, and serum levels of total bilirubin, direct bilirubin, aspartate aminotransferase, alanine aminotransferase, lactate dehydrogenase, gamma-glutamyl transpeptidase, alkaline phosphatase, albumin, blood urea nitrogen, creatinine, CRP, sodium, potassium, and chlorine; tumor factors, including the size and number of ICC, serum levels of CEA and CA19-9, gross classification of ICC, and clinical staging (tumor–node–metastasis classification) based on the criteria of the Liver Cancer Study Group of Japan (stage I, n = 0; stage II, n = 6; stage III, n = 4; stage IVA, n = 5; stage IVB, n = 16); and treatment factors such as the selected treatment modality (systemic chemotherapy, HAIC) (Table 1). The characteristics of the excluded patients are shown in Appendix A.

### 2.6. Treatment for ICC

According to the Japan Hepatobiliary and Pancreatic Surgery Society Biliary Cancer Clinic Guidelines (http://www.jshbps.jp/modules/en/index.php?cat_id=4, accessed on 17 September 2019) and the Guidelines for the Diagnosis and Management of ICC [3], systemic chemotherapy was selected as a first-line therapy for patients with unresectable ICC. For patients who did not tolerate or refused systemic chemotherapy, HAIC was selected as previously described [24,25].

### 2.7. Systemic Chemotherapy

Systemic chemotherapy was performed as previously described [26,27,28]. Briefly, in cisplatin–gemcitabine therapy, each cycle comprised cisplatin (25 mg per square meter of body-surface area) followed by gemcitabine (1000 mg per square meter), each administered on days 1 and 8 every 3 weeks, initially for four cycles [28] (Figure 2).

### 2.8. HAIC Procedure

HAIC was performed as previously described [29]. Briefly, anticancer drugs were administered via the implanted port system. Each cycle comprised 10–30 mg of cisplatin (Nichi-Iko Pharmaceutical Company, Limited, Toyama, Japan) in 30 min followed by 1000–1250 mg of 5-fluorouracil (Kyowa, Tokyo, Japan) for 3–5 days using a pressurized drug injector. Each treatment was administered on days 1 and 8 every 3 weeks (Figure 2).

### 2.9. Main Outcomes

DCR is the response rate, which is the sum of the CR, PR, and SD.

### 2.10. Survival Period

In this study, 90.4% (28/31) of patients had died by the study censor date. The survival period was defined as the period from the initial treatment for ICC to the study censor date.

### 2.11. Follow-Up Process and Assessment of Response

All patients were followed up with enhanced computed tomography (CT) performed at 1- to 3-month intervals after the initial treatment. All CT scans were taken using more than 64 raw systems (GE Healthcare Japan, Tokyo, Japan). Scanned images were read and diagnosed by two independent radiologists and one hepatologist. The maximal local radiological response of the targeted lesions was assessed according to the Response Evaluation Criteria in Solid Tumors (RECIST 1.1) [30].

### 2.12. Safety and Complications Evaluation

Adverse events (AEs) were monitored and recorded. AEs were assessed during the treatment and follow-up periods, according to the NCI Common Terminology Criteria for Adverse Events version 4.0 [31] (https://ctep.cancer.gov, https://ctep.cancer.gov/protocolDevelopment/electronic_applications/ctc.htm#ctc_40, accessed on 17 September 2019).

### 2.13. Statistics

Data are expressed as numbers or means ± standard deviations. Relationships among multiple variables were evaluated by multivariate correlation analysis. Factors or profiles associated with the prognosis of patients with ICC were analyzed using decision tree analysis. A decision tree algorithm was constructed to reveal profiles associated with the prognosis of ICC and the DCR of ICC according to the instructions provided with the R software package (URL http://www.R-project.org/, accessed on 17 September 2019) [32]. Patients with ICC were classified into the corresponding group of the decision tree algorithm. The OS of each group was estimated using the Kaplan–Meier method, and differences in survival between the groups were analyzed using the log-rank test. All *p* values were 2-tailed, and a value < 0.05 was considered statistically significant. All statistical analyses were conducted by a biostatistician (AK).

## 3. Results

### 3.1. Characteristics of Patients with ICC

The characteristics of patients with ICC are summarized in Table 1. The median age was 71 years, and 58.1% of patients were men. The median tumor size was 74 mm, the median CA 19-9 level was 106.2 ng/mL, and 51.6% of patients were stage IVB. The median albumin level, CRP level, and NLR were 3.55 g/dL, 0.95 mg/dL, and 3.19, respectively. The therapeutic strategy was based on the Japan Hepatobiliary and Pancreatic Surgery Society Biliary Cancer Clinic Guidelines, and 38.7% (12/31) of patients were treated with cisplatin–gemcitabine therapy, 58.1% (18/31) of patients were treated with HAIC, and 3.2% (1/31) of patients were treated with heavy particle beam therapy. Among the patients treated with HAIC, 44.4% (8/18) were also treated with systemic chemotherapy using gemcitabine, TS-1, or GC therapy.

### 3.2. Therapeutic Effects

The MST of all the patients was 11.6 months (Figure 3a). Moreover, the 1- and 2-year survival rates of the patients were 47.8% and 17.0%, respectively. The therapeutic effects are shown in Figure 3b. The treatment response was assessed by RECIST 1.1, and 16.1% (5/31) of patients achieved PR, 54.8% (17/31) showed SD, and 29.1% (9/31) showed progressive disease/not evaluable. The response rate (RR) was 16.1%, and the DCR was 70.9% (Figure 3b).

### 3.3. Multivariate Correlation Analysis for Survival Time

To investigate factors correlated with OS, multivariate correlation analysis was employed. Serum albumin level was positively correlated with survival time (ρ = 0.3572, *p* = 0.0485) (Table 2). Meanwhile, the serum WBC level was negatively correlated with survival time (ρ = −0.4008, *p* = 0.0280) (Table 2).

### 3.4. Decision Tree Analysis for Prognosis

A decision tree algorithm was created by using two variables to classify three groups of subjects. The WBC level was selected as the initial split variable with an optimal cut-off of 6800/μL. When subjects showed WBC levels of 6800/μL or less, 45.1% (14/31) of subjects had a long survival time (Group 1, Figure 4). When the WBC level was more than 6800/μL, 54.9% (17/31) of the subjects had a short survival time (Figure 4). Among the subjects with a WBC level of 6800/μL or more, the CRP level was selected as the variable for the second division with an optimal cut-off of 1.08 mg/dL. Thus, 64.3% (9/14) of the subjects had a short survival time when they met the following criteria: WBC level of 6800/μL or more and a CRP level of more than 1.08 mg/dL (Group 2, Figure 4). In contrast, 35.7% (5/14) of the subjects had a survival time closer to that of Group 1 when they met the following criteria: WBC level of 6800/μL or more and a CRP level of less than 1.08 mg/dL (Group 3; Figure 4). Group 1 had a statistically significantly better MST than Group 2 and Group 3 (log-rank *p* < 0.0001, Wilcoxon *p* < 0.0001, Figure 4).

### 3.5. Decision Tree Analysis for Disease Control of ICC

A decision tree algorithm was created by using two variables to classify three groups of subjects. The worst profile was patients with a CRP level greater than 4.46 mg/dL and a DCR of 28.6% (Profile 3 in Figure 5). Meanwhile, patients in Profile 1, associated with the highest DCR, had CRP levels less than 4.46 mg/dL and treatment with HAIC. The DCR was 25% higher in Profile 1 than in Profile 2 (CRP levels less than 4.46 mg/dL and without HAIC).

## 4. Discussion

In this study, we employed data mining analysis and created a new prognostic profile for patients with ICC, which consisted of WBC and CRP levels. Furthermore, we demonstrated that HAIC was associated with better disease control in patients with ICC who had low CRP levels.

In our study, the MST of patients with unresectable ICC was 11.6 months. Meanwhile, Konstantinidis et al. reported that the MST was 30.8 months in patients with unresectable ICC treated with a combination of systemic chemotherapy and HAIC with floxuridine, mitomycin C, or gemcitabine [14]. Thus, the MST in the study by Konstantinidis et al. was longer than that of our study. A possible reason for this discrepancy is the difference in ICC stages between the two studies. In the study by Konstantinidis et al., all patients had liver confined type ICC with no distant metastasis. Meanwhile, 51.6% of patients had distant metastasis in our study. In addition, in the study by Konstantinidis et al., the MST was 12.9 months in patients with unresectable ICC with distant metastasis treated with systemic chemotherapy. Moreover, Kasai et al. reported that the MST was 14.6 months in patients with unresectable ICC treated with HAIC and 5-fluorouracil combined with subcutaneous administration of pegylated interferon α-2b in patients with advanced ICC. Accordingly, the MST of our study is comparable to that in the previous reports.

In this study, multivariate analysis was not possible owing to the small number of cases; however, we employed decision tree analysis to examine the profiles associated with DCR. As a result, CRP was identified as the initial split for DCR. Several studies have reported that high CRP levels are associated with a poor prognosis and a high prevalence of local recurrence [33,34]. CRP, an index for inflammation, has various biological activities [35,36]. CRP is reported to bind to integrin α2 and Fcγ receptor I, leading to the progression of breast cancer [35]. CRP is also reported to increase the malignant properties of pancreatic cancer cells through upregulation of the ERK/AKT/STAT3 pathway [36]. Thus, CRP may be directly involved in the progression and malignant properties of cancer cells.

In patients with ICC and a CRP level < 4.46 mg/dL, HAIC was identified as the second split for DCR. In HAIC, a highly concentrated chemotherapeutic agent is injected into the liver via the hepatic artery; the consequent concentration of the agent at the tumor site would be expected to increase antitumor effects [12]. Moreover, the prevalence of severe adverse events is lower in HAIC than in systemic chemotherapy; therefore, treatment adherence is higher in HAIC [12]. Based on these features, HAIC may be associated with a better DCR in patients with ICC.

In our study, serum albumin level and WBC level were correlated with survival time. We further performed a decision tree analysis and revealed that WBC level is the most important factor associated with survival term. NLR has previously been reported as a prognostic factor for patients with ICC [37,38]. However, we found that a higher WBC count is a poor prognostic factor. It remains unclear why WBC level—but not NLR—was detected as a prognostic factor in our study. WBCs include not only neutrophils and lymphocytes but also monocytes. Monocytes are known to be associated with the prognosis of patients with gastric, colon, and pancreatic cancers [39,40,41]. In addition, Peng et al. recently reported that the lymphocyte-to-monocyte ratio predicts early recurrence of cholangiocarcinoma [42]. Moreover, Subimerb et al. reported that the CD14 + CD16 + monocyte subpopulation found in the peripheral blood of patients with cholangiocarcinoma was associated with a poor patient prognosis [43]. Thus, a high WBC level may reflect changes in monocytes in this study. Further study will be focused on the importance of monocytes for the management of patients with ICC.

The present study suggested that the inflammatory serum markers predicted the prognosis of the patients with ICC. Selection of the patients with poor prognosis is important to establish appropriate therapeutic strategies in advanced cancers. Currently, treatment options for unresectable ICC are limited. Therefore, when systemic chemotherapy such as GC therapy fails, treatment options are limited. This study and others in the literature suggest that treatments such as HAIC, radiation therapy, and TACE may also have an important role to play [10]. An investigational agent such as futibatinib may also be an option depending on future results [15], and multidisciplinary treatment may become necessary for patients with unresectable ICC.

The main limitations of this study are its retrospective nature, the single institutional experience, and small sample size with the only Asian population. In addition, the changes of the indexes such as CRP and white blood cell count fluctuate and are subject to the patient’s basic condition and the combination of tumor infection and other factors. Another limitation is the lack of multivariate analysis owing to the small number of cases, similar to previous studies [44]. We enrolled only 31 patients with ICC in this study. The sample size of this study was calculated to be 80 with a two-sided alpha of 5%, a power of 80% as previously described [45]. The number of enrolled patients was less than the required sample number; however, the incidence of ICC accounts for 4.4% of all liver tumors, and it is difficult to enroll a large number of patients in this study. In order to solve these issues, a prospective multicenter study is required.

## 5. Conclusions

In conclusion, we demonstrated a new prognostic profile for patients with ICC, which consisted of WBC and CRP levels. In addition, we demonstrated that HAIC may contribute to better disease control in patients with ICC with low CRP levels. Thus, these new profiles may be useful for the management of patients with ICC.

## Figures and Tables

**Figure 1 jcm-10-00987-f001:**
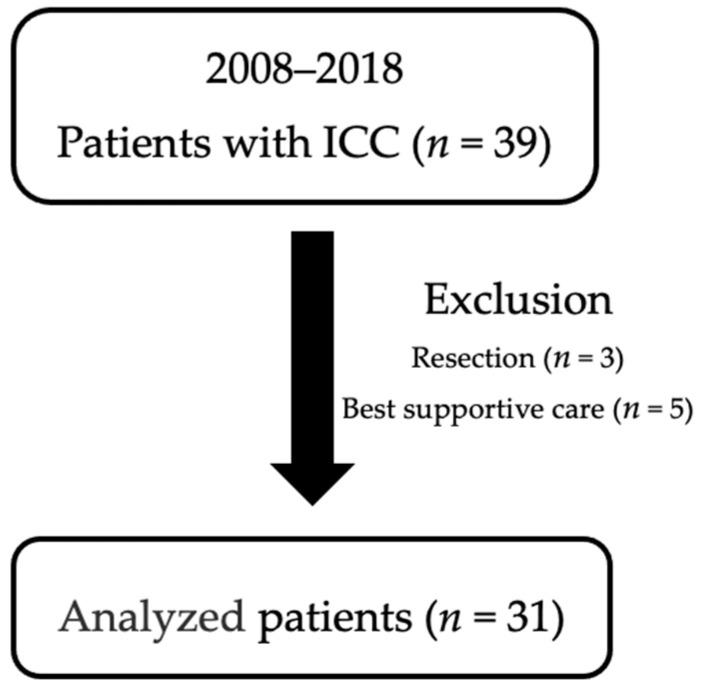
Enrolled subjects.

**Figure 2 jcm-10-00987-f002:**
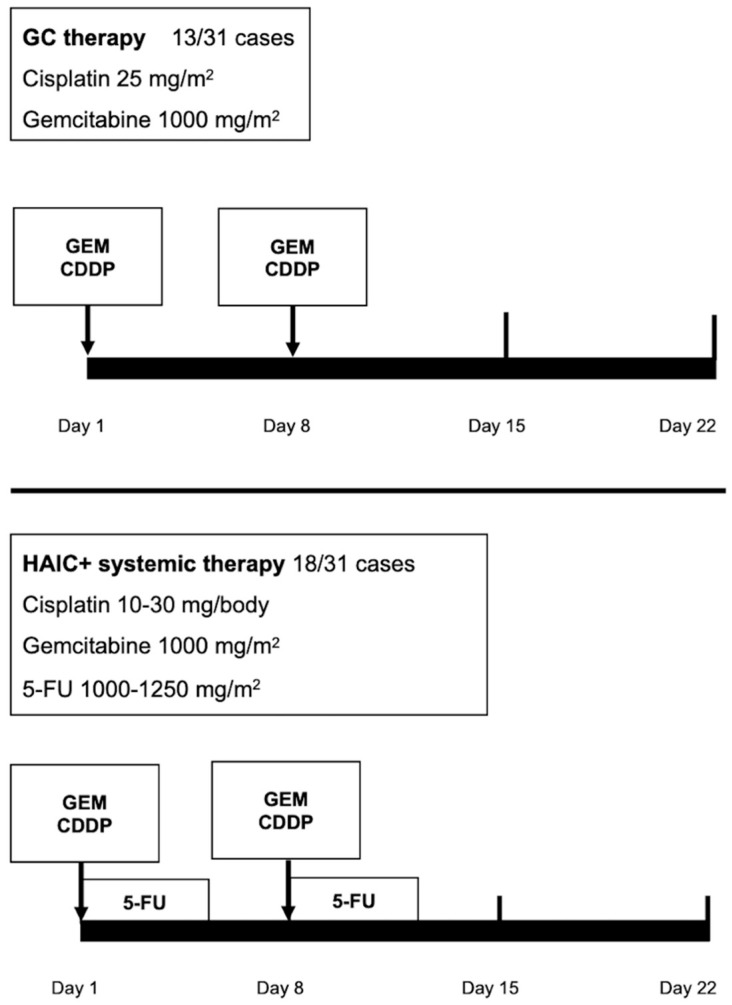
Treatment regimens for intrahepatic cholangiocarcinoma (ICC).

**Figure 3 jcm-10-00987-f003:**
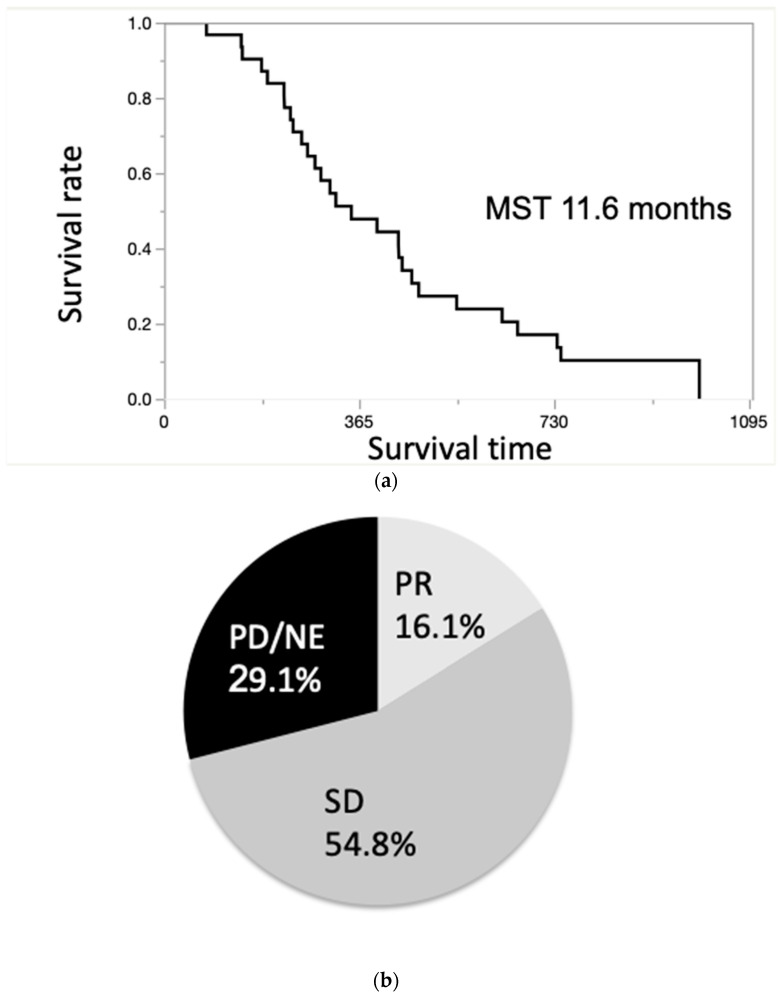
(**a**). Overall survival curve of patients with unresectable ICC. The median survival time (MST) of all the patients was 11.6 months. (**b**). Treatment effect evaluated by Response Evaluation Criteria in Solid Tumors (RECIST). The response rate (RR) was 16.1%, and the disease control rate (DCR) was 70.9%.

**Figure 4 jcm-10-00987-f004:**
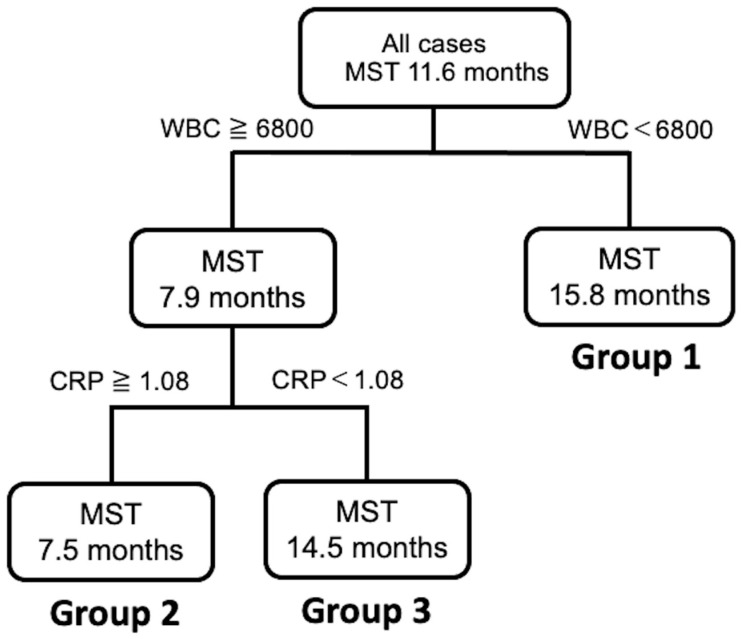
Decision tree analysis for the prognosis of patients with unresectable ICC.

**Figure 5 jcm-10-00987-f005:**
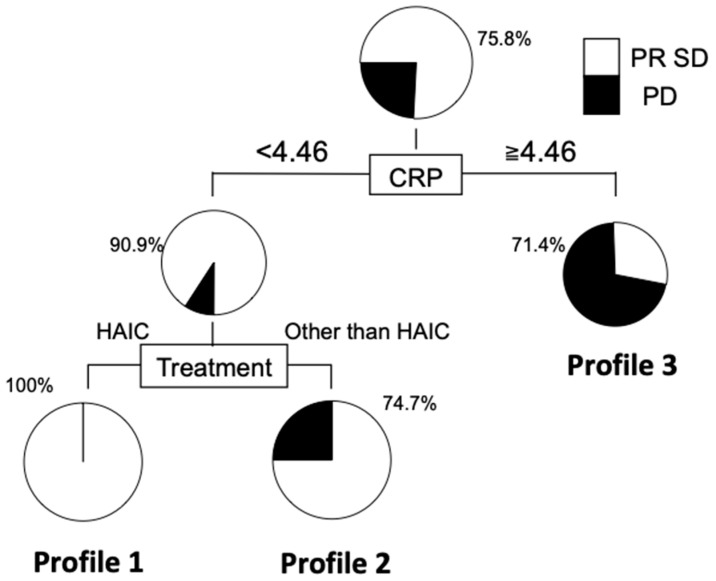
Decision tree analysis for disease control in patients with unresectable ICC. The DCR of patients with CRP less than 4.46 and HAIC was significantly higher than those without hepatic arterial infusion chemotherapy (HAIC).

**Table 1 jcm-10-00987-t001:** Baseline characteristics of patients.

Factor	Number or Median (Range)
Age (years)	71 (50–90)
ECOG PS 0/1/2	26/4/1
Sex Male/Female	18/13
Etiology HBV/HCV/non-B, non-C	3/1/27
Stage II/III/IVA/IVB	6/4/5/16
Tumor size (mm)	74 (20–197)
Total bilirubin (mg/dL)	0.92 (0.39–33.88)
AST (U/L)	48 (19–347)
ALT (U/L)	26 (11–443)
LDH (U/L)	291 (142–2511)
γ-GTP (U/L)	181 (14–3025)
ALP (U/L)	455 (240–3256)
Albumin (g/dL)	3.55 (2.49–4.36)
BUN (mg/dL)	14.4 (7.3–24.5)
Creatinine (mg/dL)	0.68 (0.41–1.12)
CRP (mg/dL)	0.95 (0.06–22)
Sodium (mEq/L)	139 (135–143)
Potassium (mEq/L)	4.1 (3.6–5.4)
Chloride (mEq/L)	104 (98–107)
Hb (g/dL)	12.9 (8.9–16.5)
White blood cells (/μL)	6300 (2600–14,600)
Neutrophils (/μL)	4718.7 (1562.6–13,484.9)
NLR	3.19 (1.82–28.58)
CEA (ng/mL)	3.4 (1.2–104.9)
CA 19-9 (U/mL)	106.2 (1–6293.6)

Abbreviations: ALP, alkaline phosphatase; ALT, alanine aminotransferase; AST, aspartate aminotransferase; BUN, blood urea nitrogen; CRP, C-reactive protein; HBV, hepatitis B virus; HCV, hepatitis C virus; LDH, lactate dehydrogenase; Hb, hemoglobin; NLR, neutrophil-to-lymphocyte ratio; WBC, white blood cell; γ-GTP, gamma-glutamyl transpeptidase.

**Table 2 jcm-10-00987-t002:** Multivariate correlation analysis of overall survival showing serum albumin level was positively correlated with survival.

	Survival Time	WBC	NLR	CRP	CEA	CA 19-9	Tumor Size	T-Bili	ALB	γ-GTP	Sodium	BUN	Creatinine
Survival time	1.0000	−0.4008	−0.2068	−0.3535	−0.1906	−0.2955	−0.0900	−0.0445	0.3572	−0.0872	0.2733	0.1841	−0.1699
WBC	−0.4008	1.0000	0.4877	0.1128	−0.0399	0.1296	0.1533	0.0332	−0.4599	0.1729	−0.3679	0.0802	−0.0499
NLR	−0.2068	0.4877	1.0000	0.2253	−0.1303	−0.1574	0.0110	−0.0817	−0.3592	−0.1277	−0.2184	0.2264	−0.0644
CRP	−0.3535	0.1128	0.2253	1.0000	0.1262	0.0921	−0.0990	0.1523	−0.1782	−0.0594	−0.1096	−0.1443	−0.0026
CEA	−0.1906	−0.0399	−0.1303	0.1262	1.0000	0.4246	−0.0804	0.0123	−0.3247	−0.0249	−0.1701	−0.0431	0.1217
CA 19-9	−0.2955	0.1296	−0.1574	0.0921	0.4246	1.0000	−0.0457	−0.1256	−0.0552	−0.0850	−0.0705	0.0032	0.0792
Tumor size	−0.0900	0.1533	0.0110	−0.0990	−0.0804	−0.0457	1.0000	−0.0994	−0.2606	−0.0711	−0.1549	−0.1808	−0.1740
T-Bili	−0.0445	0.0332	−0.0817	0.1523	0.0123	−0.1256	−0.0994	1.0000	0.2332	0.7911	0.3134	−0.2855	0.0459
ALB	0.3572	−0.4599	−0.3592	−0.1782	−0.3247	−0.0552	−0.2606	0.2332	1.0000	0.1202	0.5898	−0.0332	−0.1107
γ-GTP	−0.0872	0.1729	−0.1277	−0.0594	−0.0249	−0.0850	−0.0711	0.7911	0.1202	1.0000	0.1842	−0.0936	0.2061
Sodium	0.2733	−0.3679	−0.2184	−0.1096	−0.1701	−0.0705	−0.1549	0.3134	0.5898	0.1842	1.0000	0.1092	0.0401
BUN	0.1841	0.0802	0.2264	−0.1443	−0.0431	0.0032	−0.1808	−0.2855	−0.0332	−0.0936	0.1092	1.0000	0.0853
Creatinine	−0.1699	−0.0499	−0.0644	−0.0026	0.1217	0.0792	−0.1740	0.0459	−0.1107	0.2061	0.0401	0.0853	1.0000

Abbreviations: ALB, albumin; BUN, blood urea nitrogen; CEA, carcinoembryonic antigen; CRP, C-reactive protein; NLR, neutrophil-to-lymphocyte ratio; T-Bili, total bilirubin; WBC, white blood cell; γ-GTP, gamma-glutamyl transpeptidase.

## Data Availability

Data is contained within the article or Appendix A.

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
