# Peer review of "Therapeutic Outcomes and Prognostic Factors of Unresectable Intrahepatic Cholangiocarcinoma: A Data Mining Analysis"

_jcm, 2021, doi:10.3390/jcm10050987_

Round 1
Reviewer 1 Report
In their manuscript entitled "Therapeutic outcomes and prognostic factors of unresectable intrahepatic cholangiocarcinoma: a data mining analysis", Shirono et al. use a decision tree analysis, in order to identify profiles associated with disease control rate and of prognostic value for unresectable intrahepatic cholangiocarcinoma. The study is well designed and the results are quite interesting. In addition, the authors discuss the limitations of this study, such as the low number of patients and the origin of the diagnosis of the patients, which is from a single institution.
It would be nice if the authors could acquire access to clinical data from additional patients, in order to strengthen their findings, but I understand that this is going to be quite difficult at this point. Therefore, I will accept this study as it is.
Reviewer 2 Report
Dear Editor, thank you so much for inviting me to revise this manuscript about this interesting topic in intrahepatic cholangiocarcinoma management.
Understanding the role of these approaches in this setting is a mandatory need and the study addresses a current topic.
The manuscript is quite well written and organized. English could be improved.
Figures and tables are comprehensive and clear. However, as you could see below, some points should be elucidated.
We suggest the following modifications:
- Introduction section: although the authors correctly included important papers in this setting, we believe a couple of studies should be cited within the introduction (doi: 10.1016/j.amjsurg.2018.10.018; doi: 10.1080/13543784.2021.1837774) only for a matter of consistency. We think it might be useful to introduce the topic of this study.
- Methods and Statistical Analysis: nothing to add.
- Table 1, Baseline characteristics of study participants. The authors should report some important parameters, including GGT levels (gamma glutamyl transferase), eastern cooperative oncology group, neutrophils, hemoglobin, since some of these biochemical parameters have been suggested as independent prognostic factors for survival in biliary tract cancer patients.
- Discussion section: Interesting section.
However, some changes and some additions are necessary.
Of note, the authors should expand the Discussion section, including a more personal perspective to reflect on. For example, they could answer the following questions – in order to facilitate the understanding of this complex topic to readers: what potential does this study hold? What are the knowledge gaps and how do researchers tackle them? How do you see this area unfolding in the next 5 years?
We think it would be extremely interesting for the readers.
One additional little flaw: the authors should better explain the limitations of their work, in the last part of the Discussion.
We believe this article is suitable for publication in the journal although major revisions are needed. The main strengths of this paper are that it addresses an interesting and very timely question and provides a clear answer, with some limitations.
Certainly, the study is limited to an Asian population with very small sample size, and authors should further express this point.
In fact, it was a single-center retrospective trial and its nature should preclude the author from making strong statements.
Second, the study included a widely varied patient population from a single institute and the total number of patients analyzed was relatively small. Finally, the authors should report some baseline characteristics of patients who have not been included.
We suggest a linguistic revision, the addition of some references for a matter of consistency, and some clarifications and extensive changes regarding some crucial points in everyday clinical practice of biliary tract cancers.
Round 2
Reviewer 2 Report
The authors extensively modified the paper according to our comments.
We recommend Acceptance in the current form.